# Surgical Phase Recognition: From Public Datasets to Real-World Data

Kadir Kirtac [1,2,*], Nizamettin Aydin [2], Joël L. Lavanchy [3], Guido Beldi [3], Marco Smit [1], Michael S. Woods [1] and Florian Aspart [1,*]

1 Caresyntax GmbH, Komturstr. 18A, 12099 Berlin, Germany
2 Computer Engineering Department, Yildiz Technical University, Istanbul 34220, Turkey
3 Department of Visceral Surgery and Medicine, Inselspital, Bern University Hospital, University of Bern, 3010 Bern, Switzerland
* Correspondence: kadir.kirtac@caresyntax.com or kadir.kirtac@std.yildiz.edu.tr (K.K.); florian.aspart@caresyntax.com (F.A.)

**Abstract:** Automated recognition of surgical phases is a prerequisite for computer-assisted analysis of surgeries. The research on phase recognition has been mostly driven by publicly available datasets of laparoscopic cholecystectomy (Lap Chole) videos. Yet, videos observed in real-world settings might contain challenges, such as additional phases and longer videos, which may be missing in curated public datasets. In this work, we study (i) the possible data distribution discrepancy between videos observed in a given medical center and videos from existing public datasets, and (ii) the potential impact of this distribution difference on model development. To this end, we gathered a large, private dataset of 384 Lap Chole videos. Our dataset contained all videos, including emergency surgeries and teaching cases, recorded in a continuous time frame of five years. We observed strong differences between our dataset and the most commonly used public dataset for surgical phase recognition, Cholec80. For instance, our videos were much longer, included additional phases, and had more complex transitions between phases. We further trained and compared several state-of-the-art phase recognition models on our dataset. The models' performances greatly varied across surgical phases and videos. In particular, our results highlighted the challenge of recognizing extremely under-represented phases (usually missing in public datasets); the major phases were recognized with at least 76 percent recall. Overall, our results highlighted the need to better understand the distribution of the video data phase recognition models are trained on.

**Keywords:** Laparoscopic videos; cholecystectomy; deep learning; convolutional neural network; phase recognition; surgical data science

## 1. Introduction

Over the past decades, laparoscopic surgeries have been increasingly preferred over open surgeries. They present a shorter patient recovery time and a lower complication rate [1]. During laparoscopies, a few small incisions are made in the abdomen; surgeons are guided by the video captured with the *laparoscope*, a small camera inserted through one of these incisions. The availability of this camera recording opened the door to computer vision-based use cases such as intra-operative surgeon assistance [2] or post-operative analysis of the surgery [3,4]. Post-operative surgery analysis can be used to review the given milestone of surgery or perform post-operative skill assessment [5,6]. In the future, operating rooms (OR) will probably rely heavily on data science for context-aware assistance, and surgical phase recognition is an important step to achieving this [7]. Specifically, machine learning models have been used to automatically recognize surgical phases [8–11], i.e., to assign each frame of a surgical video to a predefined phase category. This automatic recognition of surgical phases is a pre-requisite to performing post-operative analysis at scale. For example, automated phase identification might help index large databases of

videos without manual burden. Such a system would enable faster offline video browsing or comparing surgical workflow across interventions.

Previous work on surgical phase recognition typically used a two-stage modeling approach. First, a convolutional neural network (CNN) is used to extract features from single frames, then a temporal model is applied to the extracted features (see Garrow et al. [12] for a review). These temporal models include hidden Markov models [8], LSTM networks [10,11,13], temporal convolutional networks (TCN) [9], and attention models [14,15]. Since surgery is a well-defined sequential process, the phase in the current time step is closely related to past phases and events. This is particularly challenging for surgical videos due to the excessive length of these videos compared to videos in standard action recognition datasets [16]. Therefore, methods using temporal models such as LSTMs [17,18], transformers [15], and TCNs [9] all aim to capture the long-term relation between current and past/future time steps to make more reliable predictions.

Research and development in surgical phase recognition has been mainly driven by publicly available datasets (see Garrow et al. [12] for an exhaustive list of public datasets used in surgical AI). Of these datasets, Cholec80 [8] is the most commonly referenced for surgical phase recognition. Cholec80 comprises 80 Lap Chole videos. Cholec80 was curated on purpose: videos containing alternative phases, i.e., phases which do not happen regularly, were excluded. Hence, videos in Cholec80 present mostly a linear workflow, i.e., starting with the preparation phase and ending with the retraction of the gallbladder. However, emergency surgeries can include additional phases such as *gallbladder drainage*, *liver biopsy*, and *intraoperative cholangiography*. Moreover, depending on case difficulty, these surgeries may sometimes be converted to open surgery and proceed with additional phases. Consequently, the surgical phases present in Cholec80 may not be fully representative of the surgical phase distribution observed in real-world deployment.

In the present work, we investigate the potential distribution shift in surgical videos observed in a given medical center compared to the one curated in the publicly available dataset, Cholec80. To this end, we gathered a new private dataset of 384 Lap Chole videos for surgical phase recognition. We included in our dataset all videos, including emergency surgeries and teaching cases, recorded in a continuous time frame of five years at the Inselspital.

We carefully analyzed the characteristics of our dataset and compared it with the Cholec80 dataset. For instance, we observed the presence of surgical phases such as *gallbladder drainage*, *liver biopsy*, and *intraoperative cholangiography*. These phases did not appear in other datasets before (see Figure 1 for a qualitative example).

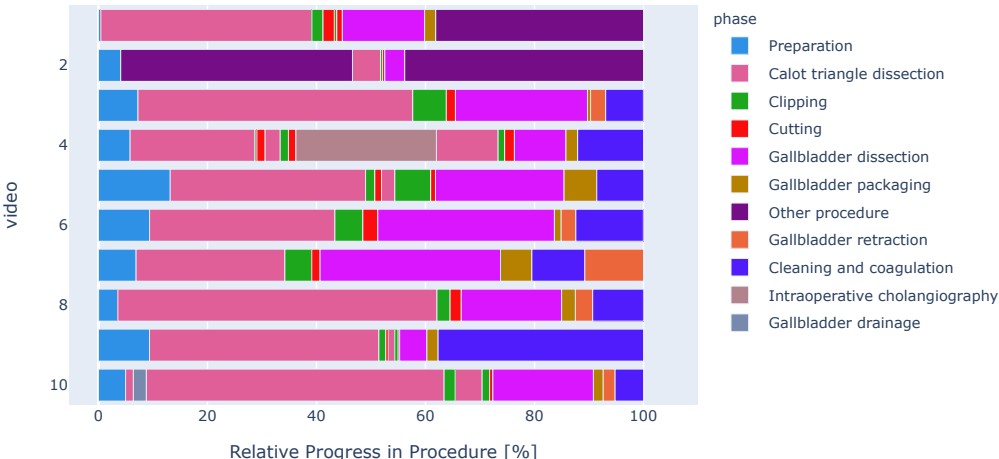

**Figure 1.** Lengths and order of surgical phases are highly imbalanced in our dataset. Color-coded ribbons illustrate phase progression for ten different procedures. Relative operation duration is shown (normalized to 100%).

Furthermore, we assessed the performance of several state-of-the-art phase detection models on this closer-to-real-world distribution. Finally, we performed an exhaustive model performance analysis on a selected model and pinpointed the challenge of severe class imbalance for a few phases.

## 2. Methods

In this study we successively (i) construct a large dataset of Lap Chole videos with our partners at Inselspital; (ii) perform a comparative analysis of our curated dataset with Cholec80; (iii) train two state-of-the-art phase recognition models on our dataset and Cholec80; (iv) finally, evaluate and compare the performance of these models on our dataset and Cholec80, using a wide variety of metrics. Each step is detailed in the following sections.

### 2.1. Ethical Approval

The institutional review board—the ethics committee of the Canton of Bern—approved the use of laparoscopic videos and waived the need to obtain informed consent.

### 2.2. Datasets

We curated a large private dataset that comprises 384 videos provided by our partners at the Department of Visceral Surgery and Medicine at Inselspital in Bern, Switzerland. During video collection, the institutional video archive was screened for video recordings of laparoscopic cholecystectomies performed between January 2014 and May 2019. All possible video recordings, including additional procedures, emergency surgery, or teaching cases, were included in our dataset.

Each video was annotated with a single surgical phase label per frame. To perform phase annotations, we adapted the guidelines provided by Endovis Surgical Workflow and Skill Analysis sub-challenge 2019 [19]. We defined additional phases, such as *gallbladder drainage*, *liver biopsy*, and *intraoperative cholangiography*, which were missing in Endovis guidelines but present in our dataset. Annotations were completed in three rounds: (1) videos were shared among 5 annotators to complete the first round of annotations, (2) a cross-check was performed to raise indecisive cases, and (3) indecisive cases were resolved in a final review round. The complete set of our annotation guidelines is illustrated in Supplementary Table S1 online.

For model training, we randomly partitioned our dataset with 70/10/20 percent training/validation/test ratios, respectively. Partitioning was performed on the video level, i.e., frames from a single video only appeared in one of the partitions. The training, validation, and test splits included 269, 38, and 77 videos, respectively.

### 2.3. Phase Recognition Models

We consider two different architectures in this work, MSTCN-based [16] and a recent transformer-based architecture, Trans-SVNet [15].

These architectures are composed of a backbone for extracting features from single frames, and a temporal model which combines features from consecutive frames to predict the surgical phase for each frame.

The feature extraction backbone is based on the Resnet-50 [20] architecture. The features extracted from the final Resnet block yield a vector of 2048 dimensions for each frame. Per-frame extracted features of a video are stacked to form a sequence, which is then fed into the temporal model.

As a temporal model, we consider (i) the online trained MSTCN model, which was named TeCNO [9], and (ii) the offline trained MSTCN model, which we named TeCNO-Off in this work.

MSTCN consists in cascading multiple stages of temporal convolutional networks (TCNs) [21]. This cascade prediction scheme allows every stage to refine the predictions of the previous stage. Every stage contains multiple layers of TCNs, which use dilated

convolutions and skip connections. Dilated convolutions offer a larger temporal receptive field with fewer parameters. Hence, this results in faster training and inference, which helps process long surgical videos.

While the TeCNO online model performs causal convolutions in its TCN layers, whereas its offline version, TeCNO-Off, performs acausal convolutions. In other words, to predict the phase for each frame, the offline version considers all surrounding frames (preceding and succeeding), while the online model considers only preceding frames.

Additionally, we considered Trans-SVNet [15], which fuses the temporal features extracted from MSTCN and spatial features extracted from Resnet-50 in its transformer layers. This spatial and temporal information fusion yielded top results on Cholec80 dataset [15]. We trained two versions of this model, one with cross-entropy loss and one with weighted cross-entropy loss (see details in Section 2.3.1).

### 2.3.1. Model Training

Videos were processed at 1 frame per second. Backbone Resnet-50 model was pre-trained on ImageNet [22], and fine-tuned on our dataset. Channel-wise image statistics, for normalization purposes, were computed on our dataset. Video frames were downsized to $224 \times 224$ pixels and data augmentation including shifting, scaling, and rotation was applied. For fine-tuning the backbone, the learning rate was $5 \times 10^{-4}$ and the batch size was 64 frames. Early stopping with a patience of 7 epochs on the validation loss was used to stop the training. In order to tackle the class imbalance inherent to different phase lengths, we used weighted cross-entropy loss. The weights were set to the inverse sample count of each phase.

The number of stages was set to 3 both for the (i) TeCNO and (ii) TeCNO-Off models, which have been trained on top of the extracted Resnet-50 features. Each MSTCN stage included 15 layers of TCNs and each layer output 64 feature maps. The models operate on the full temporal resolution of the videos. Both models were trained for a maximum of 50 epochs, with a learning rate of $5 \times 10^{-4}$. Training batch size was equal to the length of the video (one video per batch). Early stopping with a patience of 10 epochs on the validation loss was used to stop the training.

Next, we trained the Trans-SVNet model, (iii) with cross-entropy loss as described in Gao et al. [15], and (iv) with weighted cross-entropy loss. In the following, we reference the latter as Trans-SVNet-WCE. Trans-SVNet-WCE used the same class weights computed for the backbone Resnet model.

Both Trans-SVNet and Trans-SVNet-WCE used features from the same Resnet and MSTCN backbones described above. Both models had two transformer layers, each having eight attention heads. Due to memory limitations, the causal temporal sequence length could not be set higher than 15.

Models were trained for a maximum of 30 epochs, with a learning rate of $1 \times 10^{-3}$. The batch size was equal to the length of the video (one video per training batch). Early stopping with a patience of 10 epochs on the validation loss was used to stop the training.

### 2.4. Evaluation Methodology

We report the frame-wise performance and compare the metrics on the test split of our dataset. Accuracy, precision, and recall have been used previously for phase recognition [8]. Precision, recall, and f1-score are first computed per phase for each video and then averaged across videos for each phase. Finally, phase-wise scores are averaged to obtain a single score. For these metrics, the reported mean $\pm$ std correspond to variations over phases. On the contrary, we first compute the accuracy per video and then report its average across videos. For accuracy, the reported mean $\pm$ std correspond to variations over videos.

These frame-wise metrics fail to evaluate the phase segmentation performance of the model. Therefore, we additionally report segment-based metrics, e.g., metrics computed over segments of predictions and ground truths. Specifically, we incorporate the use of segmental f1-score [21] and segmental edit score [23], which have been used to assess video

action detection and segmentation algorithms [21], and lately considered for surgical phase recognition [24].

The segmental edit score penalizes out-of-order predictions while still allowing minor temporal shifts between the prediction and ground truth. For each frame-wise ground-truth sequence $G$, the segment-wise labels $G_s$ are defined such that if $G = \{[AAA], [BB], [CCC]\}$ then $G_s = \{ABC\}$, where $A, B, C$ are different labels. The predicted frame-wise ($P$) and segment-wise ($P_s$) sequences are defined analogously. The edit score is defined using an edit distance [25], $s_e(G_s, P_s)$, with insertions, deletions, and replacements. Then, the score is normalized by taking the maximum between the length of $G_s$, $L_G$, and of $P_s$, $L_P$. It can be formally defined as:

$$S_{edit} = (1 - \frac{s_e(G_s, P_s)}{max(L_G, L_P)}) \times 100 \tag{1}$$

with 100 being the best and 0 being the worst score.

The segmental f1-score measures the overlap between ground truth and predicted segments, thereby penalizing over-segmentation errors. True and false positives are computed over each predicted segment, by comparing its intersection over union (IoU) with respect to the corresponding ground truth using a fixed threshold, $k$. Precision and recall are computed for each class and then summed over classes. The segmental f1-score can be formally defined as follows,

$$F1@k = 2 \times (\frac{precision \times recall}{precision + recall}) \times 100, \tag{2}$$

where $k \in \{25, 50, 75\}$ is the IoU threshold.

## 3. Results

In the following, we first present the results of our dataset labeling. We perform a comparative analysis of our dataset with the most commonly used public dataset for surgical phase recognition: Cholec80 [8]. Finally, we thoroughly analyze the performance of our phase recognition pipeline on our dataset.

### 3.1. Comparative Analysis of Cholec80 and Our Dataset

Surgeries in our dataset were much longer than surgeries in Cholec80. Our surgeries had a median duration of 54 minutes (min: 13, max: 374) against 34 minutes (min: 12, max: 99) for the Cholec80 videos (see Supplementary Figure S1 online for a more detailed distribution of the video lengths).

Attentively inspecting each video in our dataset, we found some surgical phases which were not present in our guidelines (see Supplementary Table S1 online for the complete set of our annotation guidelines) and in the public Cholec80 dataset. In collaboration with a board certified surgeon, we identified these phases as *gallbladder drainage, liver biopsy,* and *intraoperative cholangiography*:

- *Gallbladder drainage*: a metal needle is used to puncture the gallbladder before the dissection of Calot's triangle. The aim is to reduce the inflamed gallbladder volume to avoid the spread of the gall into the abdomen;
- *Liver biopsy*: tissue samples are collected from the liver, i.e., generally towards the end of the procedures;
- *Intraoperative cholangiography*: The cystic duct is cut and a plastic tube is inserted to apply a contrast medium for cholangiography. This is an X-ray imaging of the biliary tract to check whether gallstones block the common bile duct and/or the biliary tract is dilated.

Qualitative examples of video frames associated with these surgical phases are displayed in Figure 2a–c. These three phases appeared in relatively fewer videos compared to other phases: *gallbladder drainage* in 26 videos, *liver biopsy* in 18 videos, and *intraoperative cholangiography* in 47 of all 384 videos (see Supplementary Figure S2 for more details).

Moreover, we observed less visual variability in the surgical instruments used across interventions associated with the *gallbladder drainage*, *liver biopsy*, and *intraoperative cholangiography* phases in our dataset.

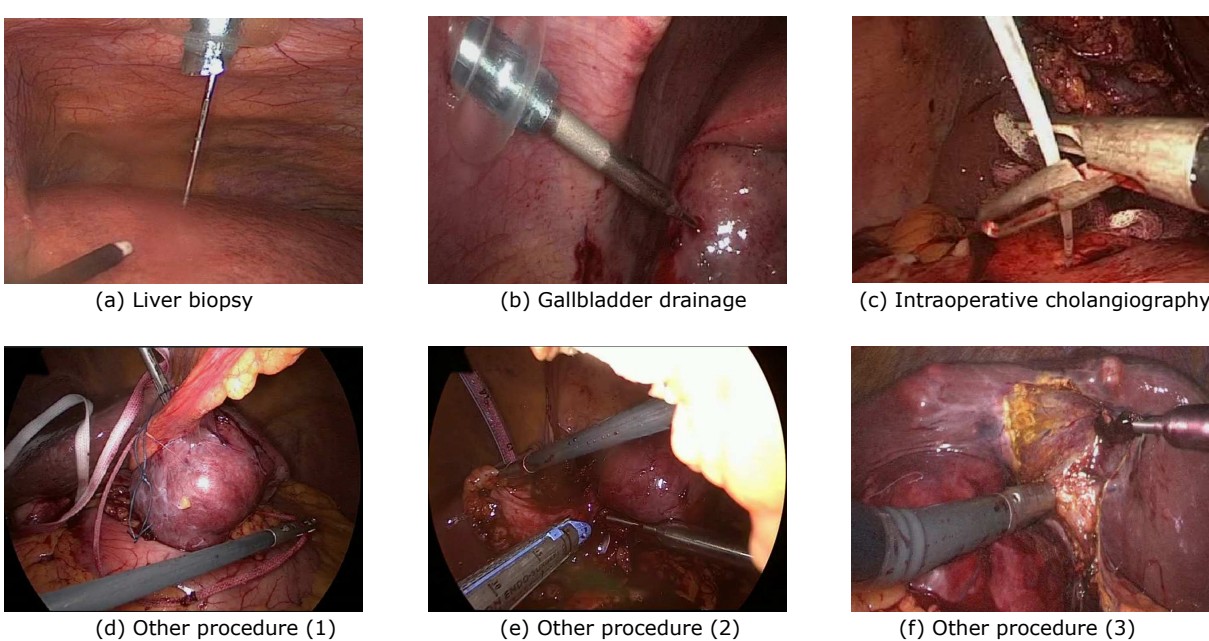

(a) Liver biopsy

(b) Gallbladder drainage

(c) Intraoperative cholangiography

(d) Other procedure (1)

(e) Other procedure (2)

(f) Other procedure (3)

**Figure 2.** Example of surgical scenes for surgical phases (from our dataset) which are absent in Cholec80: (**a**) liver biopsy; (**b**) gallbladder drainage; (**c**) intraoperative cholangiography; (**d**) application of a cotton tape for laparoscopic Pringle maneuver; (**e**) application of the endoscopic linear stapler; (**f**) intraoperative ultrasonography. These latter three examples include procedures that are not typical for cholecystectomy.

In addition to these phases, we observed the presence of non-cholecystectomy-related surgical procedures. These included procedures such as intraoperative ultrasonography and liver or pancreas resection (see Figure 2d–f for qualitative examples).

To avoid mixing these miscellaneous procedures with cholecystectomy-specific phases, we explicitly labeled these procedures as *other procedure*. We observed these non-cholecystectomy related procedures in 68 of the 384 videos we gathered.

With a median duration of 82 minutes (min: 1.2, max: 263), this phase was one of the longest in our dataset (Figure 3). Due to the variability of the tasks involved, the surgical instruments used during this *other procedure* phase varied across the interventions.

Videos in the Cholec80 dataset did not include any of the *gallbladder drainage*, *liver biopsy*, *other procedure* and *intraoperative cholangiography* phases. Consequently, Cholec80 did not include any definition for these phases.

Carefully looking at the clipping and cutting scenes, we observed that the surgical flow between these phases was not necessarily linear. We found many cases where *clipping* and *cutting* were performed multiple times; the surgeons came back and forth between the two. To enable our annotation to capture this non-linear workflow, we chose to define *clipping* and *cutting* as two separate phases. Cholec80 had a unified *clipping and cutting* phase instead.

Surgical phase duration was strongly imbalanced in both datasets (Figure 3). Besides non-cholecystectomy procedures (*other procedure*), *calot triangle dissection* was the longest phase in both datasets. In accordance with the full-video length, Cholec80 had shorter phase durations across videos and shorter video durations compared to our dataset (see Supplementary Figures S1 and S3 online).

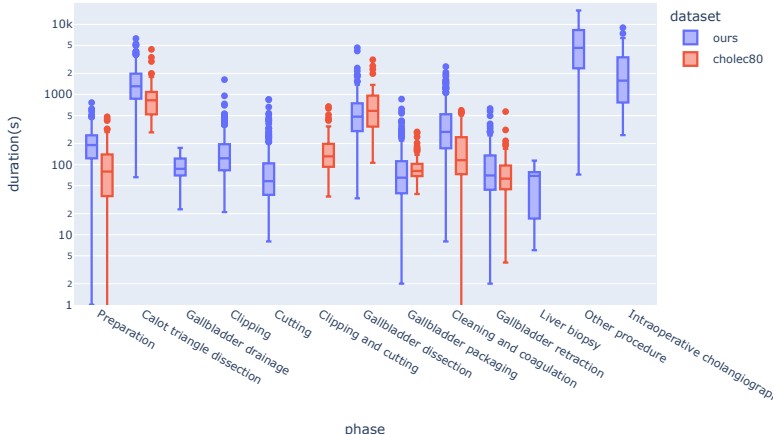

**Figure 3.** Distribution of phase duration across videos (in seconds) in our dataset and Cholec80. Duration (*y*-axis) is in log-scale. Some phases present in our videos are missing in Cholec80. Phases missing in a video are ignored.

Finally, we compared the phase transitions in both datasets (Tables 1 and 2). This table shows in what proportion (percent) a transition from a phase in the row-axis to another phase in the column-axis happens.

The surgical workflow in the Cholec80 dataset was mostly linear. Each phase had a clear following phase: four out of the six surgical phases had the same following phase in over 99% of the cases.

On the contrary, the transitions of surgical phases in our dataset were less deterministic. This was particularly the case for *other procedure* and *intraoperative cholangiography* which did not have a clear succeeding phase. Transitions to the *other procedure* phase could take place from diverse phases. In other words, the *other procedure* had no clear position in the Lap Chole workflow.

Interestingly, procedures in our dataset mostly transitioned to *cleaning and coagulation* after *gallbladder retraction*, meaning that surgeons preferred to do the cleaning after the specimen is pulled out of the body (Table 2). This was different in Cholec80, in which procedures regularly end with the *gallbladder retraction* phase (Table 1).

**Table 1.** Phase transitions are more heterogeneous in our dataset compared to Cholec80. Phase transitions in Cholec80. The numbers indicate the relative (in percent) transitions from one phase (row-axis) to another (column-axis).

| Phase | Next Phase (%) | | | | | | |
| --- | --- | --- | --- | --- | --- | --- | --- |
| | Preparation | Calot triangle dissection | Clipping and cutting | Gallbladder dissection | Gallbladder packaging | Cleaning and coagulation | Gallbladder retraction |
| Preparation | - | 100 | 0 | 0 | 0 | 0 | 0 |
| Calot triangle dissection | 0 | - | 100 | 0 | 0 | 0 | 0 |
| Clipping and cutting | 0 | 0 | - | 100 | 0 | 0 | 0 |
| Gallbladder dissection | 0 | 0 | 0 | - | 85 | 15 | 0 |
| Gallbladder packaging | 0 | 0 | 0 | 0 | - | 73.75 | 26.25 |
| Cleaning and coagulation | 0 | 0 | 0 | 0 | 17 | - | 83 |
| Gallbladder retraction | 0 | 0 | 0 | 0 | 0 | 0 | - |

**Table 2.** Phase transitions are more heterogeneous in our dataset compared to Cholec80. Phase transitions in our dataset. The numbers indicate the relative (in percent) transitions from one phase (row-axis) to another (column-axis).

| | Next Phase (%) | | | | | | | | | | | |
|---|---|---|---|---|---|---|---|---|---|---|---|---|
| Phase | Preparation | Calot triangle dissection | Gallbladder drainage | Clipping | Cutting | Gallbladder dissection | Gallbladder packaging | Cleaning&coagulation | Gallbladder retraction | Liver biopsy | Other procedure | Intraop. cholangiography |
| Preparation | - | **94.44** | 0 | 0 | 0 | 0 | 0 | 0 | 0 | 0 | 0 | 5.56 |
| Calot triangle diss. | 0 | - | 4.42 | **85.84** | 1.77 | 1.77 | 0 | 0.17 | 0 | 0.53 | 1.78 | 3.71 |
| Gallbladder drainage | 0 | **96.15** | - | 0 | 0 | 0 | 3.85 | 0 | 0 | 0 | 0 | 0 |
| Clipping | 0 | 8.38 | 0 | - | **86.93** | 2.9 | 0 | 0.16 | 0 | 0 | 0 | 1.61 |
| Cutting | 0 | 15.85 | 0 | 14.59 | - | **62.52** | 1.26 | 1.08 | 0 | 0 | 0.9 | 3.78 |
| Gallbladder diss. | 0 | 0.52 | 0 | 4.17 | 0.26 | - | **67.36** | 22.19 | 0.52 | 0 | 4.17 | 0.78 |
| Gallbladder packaging | 0 | 0 | 0.31 | 0 | 0 | 0 | - | **69.4** | 27.44 | 1.26 | 1.57 | 0 |
| Cleaning&coagulation | 0.59 | 3.55 | 0 | 1.18 | 0 | 1.18 | **47.93** | - | 13.61 | 10.65 | 21.3 | 0 |
| Gallbladder retraction | 0 | 0 | 0 | 0 | 0 | 1.26 | 0 | **96.20** | - | 0 | 2.53 | 0 |
| Liver biopsy | 0 | 0 | 0 | 0 | 0 | 0 | 0 | **96.66** | 0 | - | 3.33 | 0 |
| Other procedure | 0 | **52.72** | 0 | 5.45 | 0 | 9.09 | 3.63 | 20.0 | 0 | 7.27 | - | 1.81 |
| Intraop. cholangiography | 0 | 21.42 | 0 | **58.93** | 8.93 | 5.35 | 0 | 0 | 0 | 0 | 5.35 | - |

### 3.2. Surgical Phase Recognition on Our Dataset

We tested how existing surgical phase recognition models, developed on Cholec80, perform when trained on our dataset. Our aim was not an exhaustive comparison of all models but instead to understand how the specificity of our dataset affects the performance of existing models.

We trained state-of-the-art surgical phase recognition models on our dataset. Specifically, we considered two different architectures: MSTCN-based (TeCNO) and transformer-based (Trans-SVNet).

In accordance with existing studies on Cholec80, stacking temporal models (MSTCN or Trans-SVNet [15]) on top of Resnet-50 backbone boosted the results on all aggregated metrics (Table 3). In particular, the segment-based metric, which takes into account the temporal fragmentation of the predictions, was almost 50 times greater (*F1@50*: from 0.83 to 40.68 in the second row).

**Table 3.** Model performance (mean ± std) using aggregated metrics on our dataset. The performance of only the Resnet50 backbone is shown in the first row. Stacking the MSTCN on top of Resnet50 boosted the phase segmentation performance (the last two columns in the second row). The std is computed across the videos for *accuracy*, *F1@50* and *edit score*; across the phases for *precision*, *recall*, and *f1-score*.

| Method | Accuracy | Precision | Recall | F1-Score | F1@50 | Edit |
|---|---|---|---|---|---|---|
| Resnet-50 | 68.94 ± 13.89 | 61.54 ± 18.13 | 57.46 ± 18.18 | 52.89 ± 16.54 | 0.83 ± 1.17 | 1.79 ± 1.31 |
| TeCNO-Off | **84.75 ± 13.49** | 67.89 ± 23.23 | 69.27 ± 24.09 | 64.02 ± 22.53 | **40.68 ± 23.75** | **39.01 ± 25.3** |
| TeCNO [9] | 82.76 ± 14.20 | **69.54 ± 18.45** | **70.82 ± 20.0** | **64.65 ± 19.97** | 21.45 ± 18.28 | 18.31 ± 16.18 |
| Trans-SVNet [15] | 84.05 ± 14.41 | 69.02 ± 21.66 | 63.73 ± 30.05 | 61.70 ± 26.88 | 30.04 ± 20.13 | 26.18 ± 16.75 |
| Trans-SVNet-WCE | 82.97 ± 14.48 | 68.33 ± 22.08 | 70.67 ± 19.72 | 64.10 ± 21.95 | 26.46 ± 18.76 | 22.41 ± 15.66 |

The offline MSTCN-based model (TeCNO-Off) reached a top average accuracy of 84.75%, as well as top segment-based scores (*F1@50* and edit score). Yet, the best performance in terms of frame-wise precision, recall, and f1-score was achieved by the online TeCNO model.

Looking at the phase-wise performance of the models (Figure 4), we observed the largest variability across models on the minority phases (e.g., *gallbladder drainage*, *liver biopsy*). The performance gap between the models was the greatest in the extremely under-represented phase *liver biopsy*.

Interestingly, the online TeCNO model performed better than its offline version (TeCNO-Off) in the minority phases. This was not reflected in the aggregated accuracy metric (Table 3). Similarly, the use of weighted cross-entropy loss to train the Trans-SVNet model was crucial to increase the model recall on the minority phases (e.g., *gallbladder drainage*, *liver biopsy*). The Trans-SVNet model without weighted cross-entropy loss had a higher aggregated metric.

In the following, we performed a detailed performance analysis of the online TeCNO model. We focused on this model given its higher performance on the minority phases, the characteristic of our dataset.

The high standard deviation for precision, recall, and f1-score (Table 3) hinted at a strong variation in the model performance across phases, as observed during the model comparison (Figure 4). This variation was better reflected in the phase-wise precision/recall plots (Figure 5a).

In particular, the model struggled to accurately predict the *gallbadder drainage*, *gallbladder retraction*, *liver biopsy* and *intraoperative cholangiography*. These four phases were less prevalent in our dataset (see Supplementary Figures S2 and S3 online). Note that there were only five and four test videos for *gallbladder drainage* and *liver biopsy*, respectively. The remaining phases had much higher precision and recall. Notably, *preparation*, *Calot triangle dissection*, *gallbladder dissection*, *cleaning and coagulation* and *other procedure* phases had above 76 percent recall.

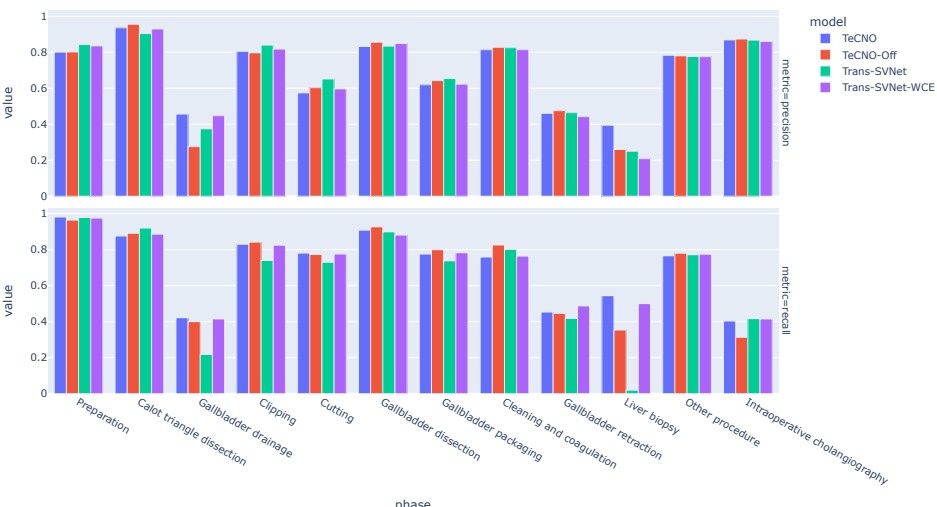

**Figure 4.** Comparative phase-specific performance analysis of the models on our dataset. For the minority phases, *gallbladder drainage*, and *liver biopsy*, the **TeCNO** [9] model showed the best performance among others. The performance gap was highly significant for *liver biopsy*, which is an extremely under-represented phase in our dataset, between TeCNO and Trans-SVNet [15] (28 times and 1.57 times greater for recall and precision, respectively).

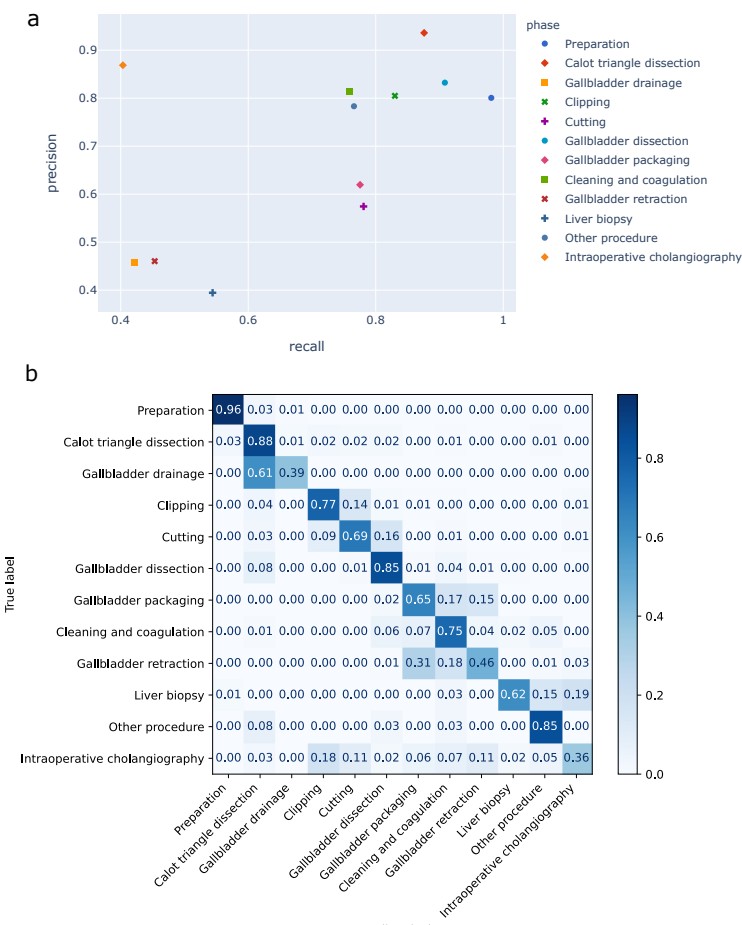

**Figure 5.** Phase-specific performance analysis of the TeCNO model on our dataset. (**a**) Illustration of phase level precision and recall averaged across videos. One can notice the strong variation in performance between phases. (**b**) The confusion matrix illustrates the confusion between phases. Every sample (frame-wise prediction and ground truth) contributed equally during the computation. Confusion between temporally closer phases was more prevalent than in temporally distant phases.

The confusion matrix emphasized the phase-specific errors of the model (Figure 5b). Importantly, confusion with temporally closer phases was more prevalent than with temporally far phases. For example, *clipping* was mostly misclassified as *cutting* and *cutting* as *gallbladder dissection*; in both cases these are their natural successive phase (Table 2). Similarly, *gallbladder packaging* was mostly misclassified as its most likely successor: *cleaning and coagulation*.

Phase-wise performance varied strongly across videos (Figure 6).

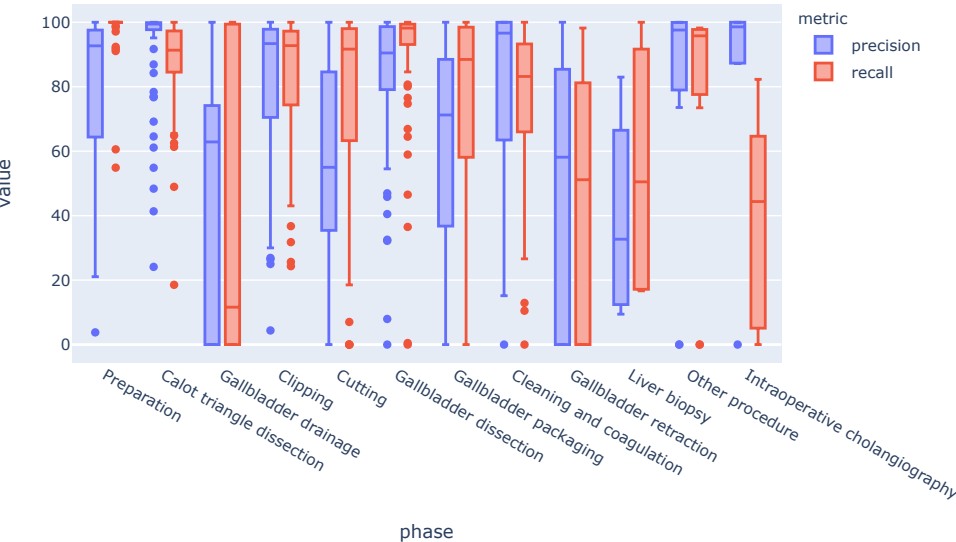

**Figure 6.** Phase-specific performance of the TeCNO model showed high variation across videos on our dataset.

While the precision/recall averaged across all videos was rather high for most of the phases, some videos displayed poor performance for particular phases. The *gallbadder drainage*, *gallbadder retraction*, *liver biopsy*, and *intraoperative cholangiography* phases had the highest variability.

Beyond frame-wise metrics, we assessed the model performance using segment-based metrics (Figure 7).

The segmental f1-score averaged over phases, *F1@k*, depended on the overlapping *IoU* threshold *k*. We measured F1@25 = 25.48 ± 18.82, F1@50 = 21.45 ± 18.28, F1@75 = 15.42 ± 13.96 and the edit score $S_{edit}$ = 18.31 ± 16.18. Similar to frame-wise metrics, the high standard deviation underlined a strong disparity in the model performance across videos.

The lower segment-based scores compared to the frame-wise f1-score emphasized an over-segmentation of the predictions. Frame-wise f1-score, segment-based edit score, and F1-score were correlated (Figure 7a). Nevertheless, segment-based scores covered a larger range than the frame-wise f1-scores. As an example, *video12* in Figure 7a had high frame-wise f1-score (84.69%), but low *F1@50* (7.36%) and edit score (6.74). Looking closer at its phase labels, this video presented two long phases, e.g., *Calot triangle dissection* and *gallbladder dissection*. These two phases were recognized with low frame-wise error rates which increased the model accuracy. Yet, predictions had lots of over-segmentation errors (fluctuating thin bars); this decreased the performance in terms of segment-based metrics.

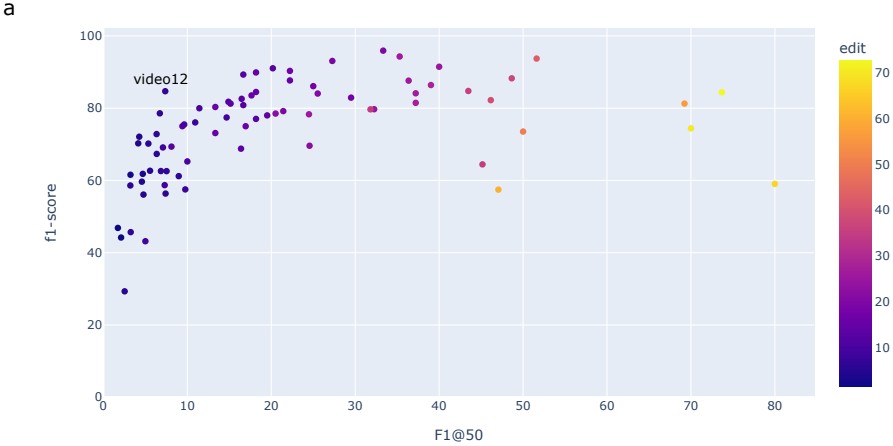

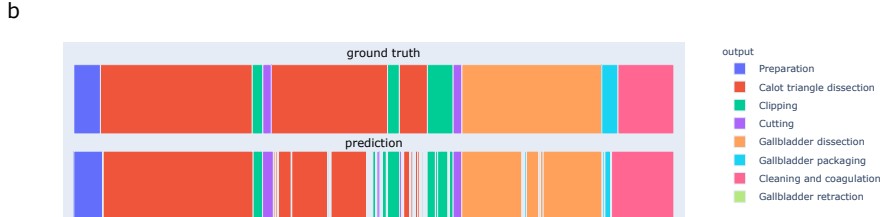

**Figure 7.** Illustration of segmental performance metrics of the online TeCNO model on our dataset, on a video basis. (**a**) Frame-wise f1-score compared to segmental f1-score and edit score. The edit score is encoded by the color bar. F1@50 displays the individual performance in a wider range compared to the frame-wise f1-score. (**b**) Timeline plot of the ground truth (top) and predictions (bottom) for *video12*, illustrating the over-segmentation errors. Each phase is encoded by a different color explained in the legend.

## 4. Discussion

In this work we thoroughly (i) analyzed the distribution of laparoscopic cholecystectomy video data recorded in a given medical center over time, and (ii) tested existing surgical phase recognition models on such video data.

We introduced a new large-scale private dataset which consisted of 384 complete laparoscopic cholecystectomy videos. Unlike curated public datasets, our dataset contained all videos, including emergency surgeries and teaching cases, recorded in a continuous time frame of five years. Consequently, we observed strong differences with the most commonly used public dataset for phase recognition: Cholec80. In particular, we observed much longer videos and the presence of additional surgical phases such as *gallbladder drainage*, *liver biopsy*, and *intraoperative cholangiography*.

Note that our dataset was gathered from a single medical center. While we considered all videos being recorded over a given timeframe, the exact distribution of the videos (e.g., the presence of specific surgical phases) may differ across other medical centers. To our knowledge, there is no study detailing the distribution of phases observed across medical centers and surgeons.

As usual in healthcare, we could not publicly share our dataset for legal reasons. We compensated for this lack of public access through an exhaustive analysis of our dataset and a thorough comparison with an existing public dataset. To our knowledge, such analysis of closer to real-world data distribution was lacking in the current literature on surgical phase recognition. Previous studies were performed on different surgery types (e.g., sleeve gastrectomy [24,26,27], right hemicolectomy [27], and appendectomy [27]) and different scale of laparoscopic cholecystectomy datasets [13,18]. The two latter studies [13,18] used private datasets larger than Cholec80, which might have included the phases presented in this article. Nevertheless, the authors focused on the model performance without detailing the difference in data distribution observed between their dataset and public ones.

We studied how existing surgical phase detection models developed on Cholec80 performed when trained on our more complex dataset. We considered two different temporal model architectures out of numerous modeling approaches for surgical phase recognition [12]. Our aim was not to make an exhaustive comparison between many models. Instead, we explored the potential impact of video data distribution on model selection. We opted for MSTCN- [16,24,28] and transformer-based temporal models for analysis, since these state-of-the-art approaches for surgical phase recognition achieved good results on Cholec80 [15].

The performances of models trained on our dataset were lower than the performance reported on Cholec80, with a margin of 7% for accuracy, and almost 20% for precision and recall. This was expected given our higher number of phases and the higher complexity of our phase distribution.

Comparing Trans-SVNet and MSTCN-based models on our dataset, the performance gap was not as significant as it was reported on the Cholec80 dataset [15]. For example, the gap between TeCNO and Trans-SVNet-WCE was less than 1% for accuracy, precision, and recall on our dataset (Table 3), compared to the 4% gap for precision and 2% gap for accuracy on Cholec80 [15]. This difference may be explained by the longer tail in our phase distribution compared to Cholec80 (Supplementary Figures S1 and S2).

The absence of some phases in individual videos in our dataset made it difficult for the temporal models to capture these phases. Consequently, Trans-SVNet [15] could not bring much improvement compared to the TeCNO model on which it was based.

The models struggled to detect less prevalent surgical phases. We attempted to tackle this imbalance by weighting the samples in the Trans-SVNet loss function according to the phase prevalence. This improved the model's capability at detecting extremely under-represented phases such as *gallbladder drainage* and *liver biopsy*. Yet, the model still displayed lower performance in minority phases.

Interestingly, the online version of MSTCN (TecNO) proved better at recognizing minority phases compared to its offline version (TeCNO-Off). This was counter-intuitive since the offline model is intrinsically able to process more contextual information: for each frame, it considers both preceding and succeeding frames. This lower performance on minority phases may be due to offline models producing smoother predictions which favors longer and more common phases.

Overall, our results shed the light on a possible model selection bias when developing models on a single publicly available dataset. The performance of surgical phase recognition has already reached a promising 90 percent accuracy [15] on Cholec80. Yet, the discrepancy between Cholec80 data distribution and data observed in more real-world scenarios might bias the model selection, e.g. when deciding the best architecture, weights, and loss function. For example, working solely on Cholec80, the developed models might become better at recognizing predominant phases but fail at capturing extremely under-represented phases.

Automation of phase recognition is a prerequisite to numerous applications ranging from post-operative analysis to intra-operative assistance. Surgeons will perform the rare surgical phases we observed in our dataset. Better understanding the data distribution in real-world scenarios is the first step to developing models that are tailored at recognizing these rare phases. While improving the accuracy of the models in the presence of rare phases, would also open the door to new applications of phase detection for these specific phases. Nevertheless, more data and dedicated modeling efforts are necessary to improve the performance for these complex phases.

## 5. Conclusions

In this work, we presented the results of the analysis of a new large-scale private dataset for phase recognition in laparoscopic cholecystectomy videos. Our analysis showed a discrepancy in the data distributions of videos in our dataset and in the most commonly used public dataset for phase recognition, Cholec80.

Additionally, we performed an exhaustive performance analysis of state-of-the-art phase recognition models on our dataset. Models struggled to recognize less prevalent surgical phases which are characteristic of our dataset.

Overall, our results highlighted the need to better understand the distribution of the video data phase that recognition models are trained on. In particular, we shed the light on the risk of model selection bias when developing models on video data that under-represents real-world scenarios.

Our future work will focus on improving the model performance in recognizing extremely under-represented surgical phases.

**Supplementary Materials:** The following supporting information can be downloaded at: https://www.mdpi.com/article/10.3390/app12178746/s1, Figure S1: Distribution of video durations (in minutes) in our dataset and Cholec80. Our dataset shows more variability in durations and have longer videos than Cholec80.; Figure S2: Distribution of phase presence as percentage in total number of videos. Some phases are extremely underrepresented across videos. For example, Gallbladder drainage and Liver biopsy are present only in 0.6% and 0.4% of videos in our dataset, respectively.; Figure S3: Distribution of phase duration as percentage of total duration in our dataset and Cholec80. The imbalance between phases for both datasets is severe. Gallbladder drainage and Liver biopsy phases are extremely underrepresented in our dataset, with 0.15% and 0.06% duration, respectively; Table S1: Definitions of phases.

**Author Contributions:** Conceptualization, K.K. and F.A.; Data curation, K.K., J.L.L. and G.B.; Formal analysis, K.K.; Investigation, K.K. and F.A.; Methodology, K.K. and F.A.; Project administration, M.S. and M.S.W.; Resources, M.S. and M.S.W.; Software, K.K.; Supervision, N.A., M.S., M.S.W. and F.A.; Validation, K.K.; Visualization, K.K.; Writing—original draft, K.K.; Writing—review and editing, K.K., J.L.L., G.B. and F.A. All authors have read and agreed to the published version of the manuscript.

**Funding:** Joël L. Lavanchy was funded by the Swiss National Science Foundation grant number P500PM_206724, accessed on 1 March 2022.

**Institutional Review Board Statement:** The institutional review board—the ethics committee of the Canton of Bern—approved the study design, the use of laparoscopic videos, and waived the need to obtain informed consent (KEK 2018-01964). All methods were performed in accordance with the relevant guidelines and regulations.

**Data Availability Statement:** The data that support the findings of this study consist of published and restricted data. Published data are available from the reported references. Restricted data are not publicly available.

**Conflicts of Interest:** The authors declare no conflict of interest.

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
