# Peer review of "Surgical Phase Recognition: From Public Datasets to Real-World Data"

_applsci, doi:10.3390/app12178746_

Round 1

Reviewer 1 Report

No comments

Author Response

We thank the reviewer for these positive feedbacks.

Reviewer 2 Report

Dear Authors,

A very interesting, well designed and written study. The novelty and research questions are well addressed. Limitations of the study are missing.

Kindly provide in discussion similar studies if exist. 

Also justify the title of the paper.

Author Response

Point 1: Limitations of the study are missing

Response 1: We thank the reviewer for this comment. The study has two main limitations:

(i) we gathered all videos from a single medical center over a given period. While we observed additional surgical phases, the exact data distribution might differ across medical centers.

(ii) We tested only two out of numerous existing phase recognition models from the literature. Our aim was not to benchmark all possible models on our dataset. Instead, we opted for a more exhaustive (e.g., phase-wise) analysis of the models performance, shedding the light on potential effect of different data distribution on the model selection.

We updated the discussion (p.9,10) to include these limitations.

Point 2: Kindly provide in discussion similar studies if exist

Response 2: As we discussed in the introduction and discussion, numerous studies exist on surgical workflow recognition, most of them working with public datasets. A few modelling studies were performed on large private dataset, yet none of these studies presented a detailed comparison between their dataset and public ones. We believe this information is particularly relevant for researchers in the field who have solely access to public, curated datasets.

We updated the discussion to better (p.9,10) discuss existing similar studies.

Point 3: Also justify the title of the paper

Response 3: In this paper, we shed a light on a data distribution divergence between existing public dataset for phase recognition and an non-curated private dataset, which is closer to real-world scenario. As we discussed, this data distribution divergence can impact the selection of phase recognition models and therefore impact their final use case. We added a corresponding discussion at the end of the Discussion section.

Reviewer 3 Report

I read with interest the article of Kirtac et al. evaluating the phase of the laparoscopic cholecystectomy comparing with different data set.
The paper showed the ongoing importance of the link between the surgery and technologies, but showed also the importance of the difference based on the dataset used. My comments: I would make more clear in the methods: In particular I would make more clear the endpoints of the study, making by point by point. In the discussion, I would add the opinion author about the role of the paper finding in the daily bases activity: 1. How a general surgeon can use these results? 2. What is the impact of these on the daily practice? 3. What is your suggestion about the management of these dateset?

Author Response

Point 1: I would make more clear in the methods: In particular I would make more clear the endpoints of the study, making by point by point.

Response 1: We thank the reviewer for this comment. We added a point by point enumeration of the study endpoint in the first paragraph of the Methods section.

Point 2: In the discussion, I would add the opinion author about the role of the paper finding in the daily bases activity: 1. How a general surgeon can use these results? 2. What is the impact of these on the daily practice? 3. What is your suggestion about the management of these dateset?

Response 2: We thank the reviewer for this comment. Our work shed the light on a data distribution divergence between public datasets and a non-curated (i.e., closer to real-world data) private dataset. Most of the phase recognition models are being developed on a public dataset. These models may not be optimized to detect unusual phases. This can impede the use of phase detection algorithm in real-world settings. Additionally, detecting such unusual surgical phase may open the door to new applications of phase recognition algorithms.
We added a paragraph at the end of the Discussion section.

Reviewer 4 Report

Thank you for your submission. The testing methodology was designed and explained well. The results are clearly organized and displayed and the conclusion is well within reason.

Author Response

We thank the reviewer for these positive comments.

Round 2

Reviewer 3 Report

Would accept as it is